# Conditional deep learning model reveals translation elongation determinants during amino acid deprivation
Mohan Vamsi Nallapareddy [1,3], Francesco Craighero [1,3], Lina Worpenberg[2], Felix Naef[2],
Cédric Gobet [2] ✉ & Pierre Vandergheynst [1] ✉

Translation elongation plays a key role in cellular homeostasis, and dysregulation of this process has been implicated in various diseases and metabolic disorders. Uncovering the causes of intragenic heterogeneity of translation, especially in contexts of different amino acid deprivations, could help increase our understanding of these disorders and pave the way for novel therapeutics. Ribosome profiling provides accurate measurements for the genome-wide ribosome footprints, which could be utilized to investigate these mechanisms. Here we present Riboclette, a conditional deep learning model featuring a dual output head that uses the mRNA sequence input to accurately predict the ribosome footprint profiles across six amino acid deprivation conditions. Exploiting standard interpretability methods, we identify specific codons related to deprived amino acids, poly-basic regions, and negatively charged amino acids as the primary drivers of the stalling response. Moreover, we use Riboclette to extract motif level drivers for ribosome stalling by performing in silico perturbation experiments. These motifs precisely explain stalling at different codon positions, allowing for the differentiation between expected determinants of rare stalling events. Our framework offers an accurate and explainable method for understanding the impact of intragenic variations on the regulation of translation elongation under amino acid deprivation.

Controlling the synthesis of proteins from messenger RNA (mRNA) through translation is essential for gene expression regulation, directly influencing protein abundance[1] and cellular homeostasis[2]. While recent advances have shed light on the dynamic regulation of translation[3,4], a precise characterization of its high intragenic heterogeneity remains elusive[5]. A better understanding of translation regulatory mechanisms would enable crucial progress in developing novel therapeutic strategies, as translation dysregulation has been linked to the progression of numerous metabolic-associated diseases such as atherosclerosis, non-alcoholic fatty liver disease[6], and even cancer[7].

Translation is composed of four phases: initiation, elongation, termination, and ribosome recycling. It is widely recognized that initiation is the primary regulator for protein synthesis[8]; however, elongation has also been recently reconsidered as another important factor[9]. Notably, it has been implicated in maintaining mRNA stability[10] and the regulation of protein function by influencing co-translational folding[11]. The introduction of ribosome profiling (Ribo-seq)[12], a sequencing technique capable of

accurately mapping the positions of translating ribosomes on mRNA, has led to the identification of key regulatory mechanisms of translation elongation, such as codon usage and codon context[13]. Despite the dynamics of elongation being well studied in unicellular organisms[14], its regulatory mechanisms in higher eukaryotes remain poorly characterized[15]. Accordingly, recent research is focused on exploring the gene-level heterogeneity of translation in mammalian cells under various amino acid deprivation conditions[16,17]. A deeper understanding of the effects of nutrient deprivation could lead to better insights into disease states[7] and potential new treatments[18].

With important advances such as the Transformer architecture[19], deep learning is increasingly becoming central to genomic research[20]. While well-established models to predict gene regulatory functions from DNA sequences already exist[21], transformer-based approaches targeting Ribo-seq data have only emerged recently[22,23]. The core task involves predicting the Ribosome Footprint Profiles (RFPs) given the coding mRNA sequence (CDS)[24], with alternative models focusing on predicting one context-

[1]Signal Processing Laboratory 2 (LTS2), IEM, STI, École Polytechnique Fédérale de Lausanne (EPFL), Lausanne, Vaud, Switzerland. [2]Institute of Bioengineering, School of Life Sciences, École Polytechnique Fédérale de Lausanne (EPFL), Lausanne, Vaud, Switzerland. [3]These authors contributed equally: Mohan Vamsi Nallapareddy, Francesco Craighero. ✉e-mail: cedric.gobet@epfl.ch; pierre.vandergheynst@epfl.ch

dependent ribosome profile from another[23], or exploiting additional modalities, such as RNA-seq[22] or the mRNA secondary structure[25] (Table S1).

In this study, we introduce Riboclette, a conditional double-headed deep learning model aimed at predicting ribosome footprint profiles from mRNA CDS under both control and amino acid-deprived conditions (Fig. 1A–D). To accurately capture the impact of the deprivations on ribosome elongation rates, Riboclette features a control head (CTRL) for predicting the control profiles and a deprivation difference head ($\Delta D$) for predicting the difference between the amino acid-deprived condition (DC) and the CTRL profiles. By predicting the deprivation difference, we aim to mitigate technical artifacts, enabling the model to focus solely on the biological factors affecting the deprivation condition. Furthermore, by interpreting the predicted deprivation difference counts using explainable AI tools[26], we can precisely identify intragenic variations under different conditions by analyzing the codons surrounding each position in the coding sequence that impact ribosome elongation in a context-dependent manner. To capture interactions between significant codons, we also perform an in silico perturbation analysis of Riboclette predictions, resulting in a set of codon motifs with a strong predicted impact on ribosome stalling. Notably, Riboclette's pipeline also employs standardized and reproducible preprocessing steps and a training process involving pseudo-labeling[27] to reduce the impact of noise on the model's performance.

Due to the high heterogeneity of translation dynamics, focusing only on global model explanations could potentially obfuscate ribosome stalling determinants that impact only specific gene subsets. For this reason, we developed a web server (see Data Availability) to enable easy navigation of the datasets, Riboclette outputs, and explanations. This platform allows users to visualize the predicted RFPs for each gene and examine the factors

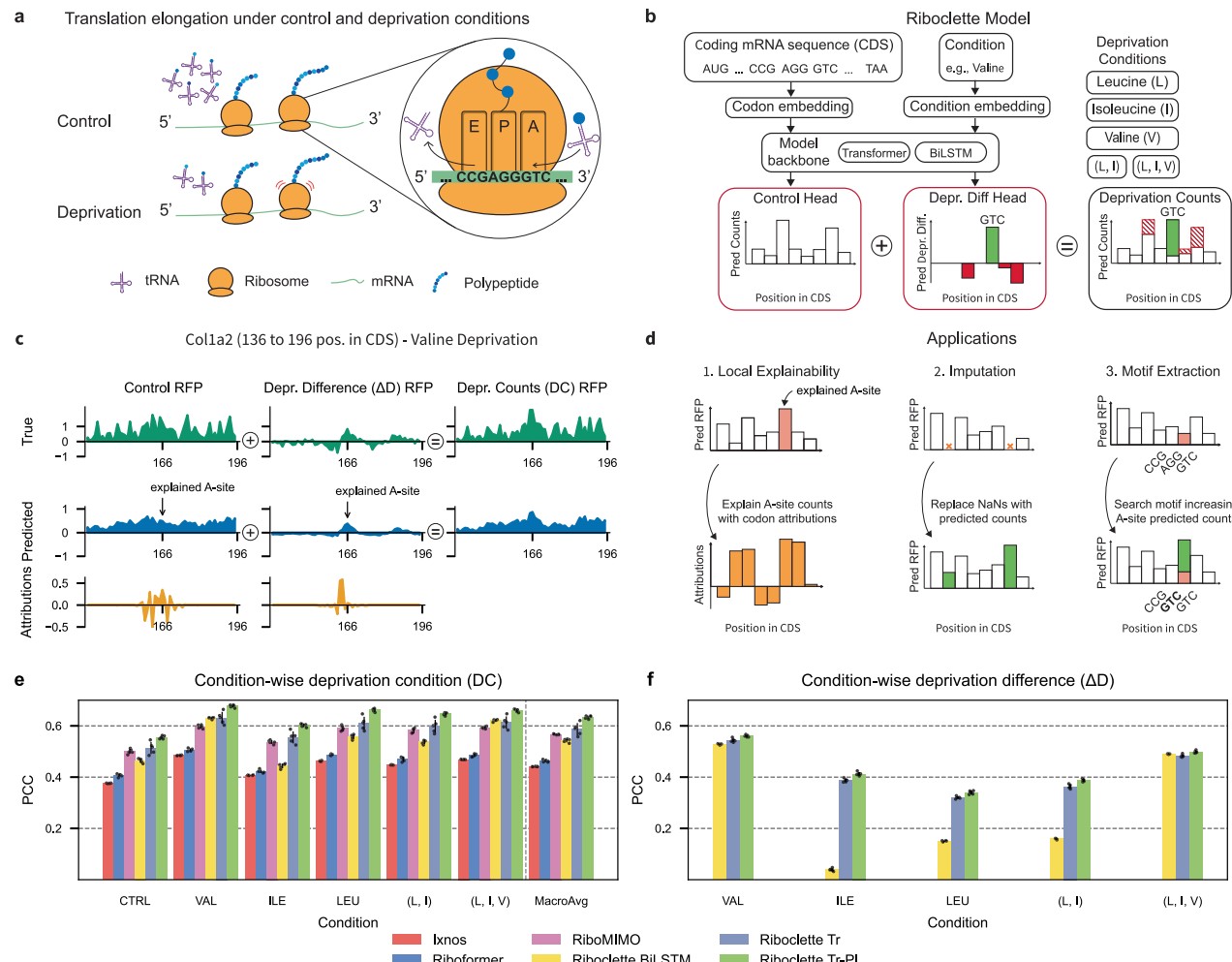

**Fig. 1 | Riboclette model task, diagram, outputs, and applications. a** Schematic of translation elongation under control and amino acid deprived condition. In the example, when GTC is at the A-site, the ribosome slows down only in the deprived condition. **b** Schematic representation of the Riboclette model which uses the mRNA sequence and the amino acid deprivation condition as the inputs to predict the control (CTRL) Ribosome Footprint Profiles (RFPs) and deprivation difference ($\Delta D$) Ribosome Footprint Profiles ($\Delta$RFPs) using the Control and Depr. Diff output heads respectively (highlighted with a red border). The outputs from both of these heads are added to obtain the deprivation condition (DC) RFPs. The proposed model is tested with five deprivation conditions: leucine, isoleucine, valine, double (leucine and isoleucine), and triple (leucine, isoleucine, and valine) in addition to the control condition. Following the example in sub-figure (a), the model predicts higher deprivation counts at GTC. **c** Example Riboclette outputs for the *Col1a2* gene under Valine deprivation from position 136 to 196 in the Coding Sequence (CDS). Here, we show true and predicted control RFPs, $\Delta D$ $\Delta$RFPs, and DC RFPs, together with the codon attributions corresponding to the selected A-site position. **d** The Riboclette model is applied for three main tasks. The first is to provide local explainability metrics via codon attributions to understand ribosome stalling at a selected A-site position. The second is to impute the missing values in the RFPs present in the dataset. Finally, the third application is to extract codon motifs to explain ribosome stalling. **e** Comparison of the Pearson Correlation Coefficient (PCC) between the predicted and true RFPs for the baseline and the three Riboclette variants across all conditions, along with their macro-average (MacroAvg). The error bars here represent the standard deviation of PCC. **f** Comparison of the PCC between the predicted and true $\Delta$RFPs for the baselines (RiboMIMO, Riboformer, and I$\chi$nos) and the three Riboclette variants across all conditions. The baselines are excluded from this analysis because they do not predict $\Delta$RFPs. The error bars here represent the standard deviation of the PCC.

influencing ribosome elongation at every codon position, enhancing the interpretability of Riboclette's decision-making process.

## Results

### Transformer-based Riboclette is the best performing model across conditions

One of the methods to study the intragenic heterogeneity of translation elongation rates is to study how different genes respond to perturbations on translational dynamics. To assess these effects, we induce distinct amino acid deprivations that might decrease specific tRNA aminoacylation levels, leading to increased ribosome dwell times and stalling. Specifically, we analyzed RFPs obtained from mouse cells under five different amino acid deprivations alongside the control condition[17]. The deprivation conditions involved a specific set of essential branched-chain amino acids (BCAAs): Leucine (LEU, L), Isoleucine (ILE, I), Valine (VAL, V), or combinations of them, that is, (L, I) and (L, I, V). Hereafter, references to these amino acids will indicate the condition in which they are deprived.

To discover the codon contexts driving the regulation of each condition, we developed a conditional double-headed deep learning model called Riboclette. By design, this model provides explanations of the effect of amino acid deprivations on RFPs. The model accepts as input the CDS of a given gene and the corresponding condition identifier, which indicates either deprived amino acid information or control condition. The Riboclette model is defined to be conditional, meaning that the predicted RFPs for a given input gene depend on the condition identifier. This is implemented by encoding both the gene's CDS, and the condition identifier into a sequence of learnable codon and condition embeddings. These learned embeddings are then fed into the Riboclette backbone, which can be any Natural Language Processing (NLP) model of choice. In this study, we consider model backbones based on both a Transformer (Tr) and a Bidirectional LSTM (BiLSTM). The backbone predicts both the gene CTRL RFPs and the difference between the deprived and the CTRL RFPs; we refer to these two prediction outputs as the control (CTRL) and deprivation difference ($\Delta D$) heads, respectively. The target deprived condition (DC) RFP then simply corresponds to the sum of the outputs from these two heads.

Several computational models have been proposed for predicting ribosome counts using the mRNA sequence. Among the earliest deep learning methods is I$\chi$nos[28], which uses a codon-window input in conjunction with a simple feed-forward network. More recent methods, such as Riboformer[23] and RiboMIMO[24], leverage advanced architectures such as multi-head attention[19] and gated recurrent units[29], respectively. Riboformer is currently the only existing deep learning method capable of dealing with conditioned outputs. However, its dependence on the starting condition RFP and the fact that it uses a codon-window input restricts its applicability. In contrast, RiboMIMO incorporates the entire mRNA CDS to predict the complete RFP. Although other methods, such as the Extended Isolation Forest (EIF)[30], and Translatomer[22] have been proposed for this task, they significantly differ in their task formulation and scope. The EIF model performs a different task compared to us, as it focuses primarily on predicting the location of stalling sites in a sequence. While Translatomer addresses the same task as ours, it uses RNA-Seq information as an additional modality, altering the scope of the study. Considering this, for baseline comparisons we employ RiboMIMO, I$\chi$nos, and a variant of Riboformer that relies solely on the mRNA CDS, to ensure a fair comparison. As these baseline models do not account for different deprivations, we trained separate instances of each model tailored to each condition.

To evaluate the performance of each model, we first pre-processed the data and filtered RFPs to populate a high-quality left-out test set. Then, to select the best hyperparameters, the remaining data was split into a train and validation set (selected hyperparameters are reported in Table S2). We evaluated the models considering the Pearson Correlation Coefficient (PCC) between the true and predicted RFPs for the final output (DC) (Figs. 1E and S2 for additional evaluation results). We notice that Riboclette Tr is the best performing model across all conditions, outperforming all

three baseline models, with a macro-average (MacroAvg) PCC of $0.587 \pm 0.024$. The lower performance of the window-based baseline models, I$\chi$nos and Riboformer, can be attributed to their limited input context, which may overlook relationships spanning the entire sequence. In contrast, models that use the full mRNA CDS, RiboMIMO and Riboclette BiLSTM, achieve performance closer to the transformer-based Riboclette, although with less consistency across conditions.

Ribosome profiling data is frequently characterized by a substantial number of missing values, such as NaNs and zero counts, stemming from challenges in mapping short reads and limitations in sequencing depth. To address this, we tested pseudo-labeling[27] as a strategy to enhance the quality of the data and, in turn, the model performance (Fig. 1E). The approach involves using Riboclette Tr, the best-performing model, to impute the missing values for all the genes in the training set, creating a pseudo-labeled dataset, while leaving the validation and test sets unchanged. This process increases the training dataset size from the original 16879 samples to 91116. A pseudo-labeled variant of Riboclette Tr, which is termed Riboclette Tr-Pl, is then trained on the larger pseudo-labeled dataset. This model attains higher performance and lower variance compared to Riboclette Tr and other baselines (MacroAvg PCC = $0.633 \pm 0.006$), as seen in the pairwise model performance comparison (see Fig. S3). In particular, the pseudo-labeled model shows an increased performance across all the conditions. This can be mainly attributed to the higher performance of the CTRL head, as the performance on $\Delta D$ only increases slightly from Riboclette Tr to Riboclette Tr-Pl (Fig. 1F). Specifically, pseudo-labeling has allowed Riboclette Tr-Pl to better handle the technical variability in the CTRL condition, which arises from the fact that this was generated by merging two distinct control datasets obtained from different cell types and library preparation methods (see Fig. S4 and "Experimental Pipeline" in the "Methods" section). Hereafter, Riboclette will refer to the best-performing Riboclette Tr-Pl model.

### Riboclette predictions are primarily affected by adjacent codons

The Integrated Gradients algorithm[26] was used as a gradient-based feature attribution approach to identify key codon positions in the input CDS for predicting the Ribosome Profiling Count (RPC) at a given output codon position (the A-site). We focused on positions exhibiting marked ribosome stalling, which were identified by RPC or $\Delta$RPC (in the case of $\Delta D$) exceeding the gene's mean plus one standard deviation (referred to as "peaks" in the RFP). For these positions, the distribution of the top-5 most salient positions for CTRL and $\Delta D$ separately (Fig. 2A) was computed. In both distributions, the key codons belong to the immediate neighborhood of the A-site mostly inside the ribosome protected fragment region (−5 to +4 codons with respect to the A-site), with the most important positions being the E, P, and A-sites.

We observe an unusually high attribution for the codons at positions −5 and +3 in the control distribution, which are adjacent to the end regions of ribosome footprints. This likely reflects technical biases from the Ribo-seq experimental pipeline, corroborating previous findings[28]. Conversely, the $\Delta D$ distribution is much cleaner, with less pronounced attributions at these positions, leaving mostly the deprivation-induced signal. Considering each condition individually, we found that only ILE exhibits residual technical bias from the CTRL condition at position −5 (Fig. S5). Moreover, codons upstream of the A-site exhibit greater influence compared to the CTRL attribution profile. Both the CTRL and $\Delta D$ global attribution profiles indicate that RPC predictions at the A-site predominantly depend on the codons within a 10-codon window upstream and downstream. This region will be referred to as the Significance Window (SW).

### Riboclette attributions identify deprivation-specific codons affecting ribosome stalling

From the analysis of the attribution values, we notice that Riboclette not only prioritizes the codons based on the proximity to the A-site but also assigns greater importance to those that are expected to show stalling in a given condition. This is supported by the correlation between codons with high

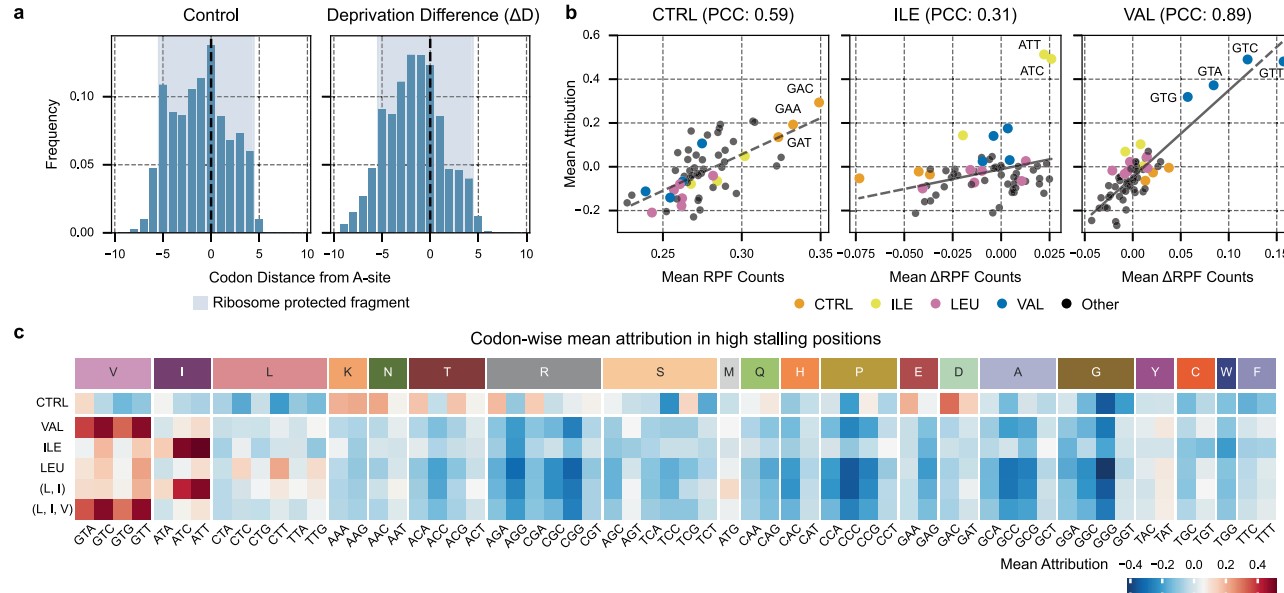

**Fig. 2 | Position and codon-wise attribution analysis.** These panels examine the attributions for high stalling codon positions across all genes where the RFP ($\Delta$RFP) counts exceed the mean plus one standard deviation of the gene's RFP ($\Delta$RFP) counts (referred to as "peaks" in the RFP). These peaks were considered to be the A-sites, and the surrounding attributions in a 10-codon window up- and downstream (the Significance Window) were max normalized and examined. For samples in the control condition, we used the attributions from the CTRL head, and for samples in an amino acid deprived condition, we used the attributions from the $\Delta D$ head. **a** Distribution of the top-5 most salient positions according to the attribution from the peak A-sites, displayed for the CTRL and $\Delta D$ heads. For both of these distributions, the ribosome protected fragment has been highlighted in blue. **b** Correlation between the codon-wise mean attributions for all the samples in the dataset and their mean RPC (or $\Delta$RPC in case of amino acid deprivation) for Control (CTRL), Isoleucine (ILE), and Valine (VAL) conditions. Similar analysis for LEU, (L, I), and (L, I, V) deprivations can be found in Fig. S6. **c** Codon-wise mean attribution across the different conditions, divided by the amino acid encoded by the codons. The mean attribution scale quantifies the codon's influence on the A-site, where negative values indicate a reduction in the expected ribosome profile count (RPC), accelerating translation, while positive values signify an increase in the expected RPC, slowing down the ribosome.

mean attribution values and the ones with high mean RPC and $\Delta$RPC, for CTRL and $\Delta D$ (Figs. 2B and S6).

Conducting the analysis on the SW around the A-site, we compare each condition by computing the codon-wise normalized mean attributions (Fig. 2C). In CTRL, the codons coding for negatively charged amino acids such as Glu (E: GAA) and Asp (D: GAC and GAT) emerge as the slowest and most important codons[31]. Additionally, we notice that poly-basic stretches with residues such as Lys (K: AAA, and AAG) and Arg (R: AGA, and CGA), which are known to be ribosome collision sites in yeast[32], have been detected by Riboclette to be considerably slow. Similar patterns in the CTRL condition were previously observed in one of the datasets reused in this study (see "Experimental Pipeline" in the "Methods" section), where GAT, GAC, and GAA were also found to be slow codons[15].

Meanwhile, in the single deprivation conditions, the salient codons correspond to the deprived ones, confirming that Riboclette accurately learned the deprivation-induced patterns. We observe an interesting synonymous codon bias where some deprived codons cause a stronger stalling effect as opposed to others. Specifically, ATC and ATT for ILE, GTC, and GTT for VAL, and CTC, CTT, and TTG for LEU. We find the LEU deprivation to have a milder effect as opposed to the other single deprivations. This observation, in addition to the three specific LEU codons identified to be driving this stalling, is supported by previous studies on LEU deprivation[16]. Overall, VAL has the strongest deprivation effect, followed by ILE, and LEU being the mildest (Fig. S13A), allowing VAL to be the easiest condition to predict. The combined deprivations mirror the amino acid leading to the strongest stalling: ILE for the double deprivation (L, I), and VAL for the triple deprivation (L, I, V), with the other deprived amino acids only influencing them weakly. These findings regarding the stalling codons in different conditions, synonymous codon bias for deprived codons, and deprivation intensities have been corroborated by an orthogonal approach[17]. Finally, as a validation for Riboclette's ability to isolate deprivation-specific determinants, we do not observe any strong correlation between salient

codons in the CTRL and deprivation conditions, with the highest codon-wise mean attribution correlation between CTRL and another condition being just 0.053.

## Riboclette perturbation analysis reveals codon motifs for ribosome stalling

Ribosome stalling patterns depend on particular codon motifs, as previously suggested in various studies in yeast[23,33]. To precisely characterize such motifs in silico, we exploit the double-head design of Riboclette to search for minimal codon mutations that maximize RPC or $\Delta$RPC for selected codon positions in the CTRL or $\Delta D$ heads, respectively. Notably, this approach is inspired by counterfactual explanations[34], which aim to find minimal input feature perturbations that alter a model's predicted output, thereby providing insights into its behavior.

For each of the six conditions, we randomly selected 1000 codon positions from the dataset where the deprivation RPCs were lower than the gene's RFP mean minus one standard deviation (referred to as "non-peak" positions) and then extracted the surrounding SW. Using a beam search (Fig. 3A), we looked for minimal mutations of up to three codons that lead to the highest predicted percentage increase of RPC or $\Delta$RPC, for CTRL and $\Delta D$, respectively. Since the initial counts could be close to zero, we overcome the issue of calculating the percentage increases by employing the MAAPE metric[35]. Specifically, we added one codon mutation at a time and cached the top 5 motifs (the beam search width) for each new perturbation, ending up with a total of 155 motifs of lengths 1–3 per codon position SW. The beam search method was chosen in contrast to a computationally expensive data-based approach, where one would have to mine the dataset for all possible gene windows and study the increase in RPC. Exploiting Riboclette greatly simplifies the analysis, isolates the technical biases, and potentially helps us discover motifs that only rarely occur in the data.

The discovered codon motifs consist of up to three codons. If these codons are not contiguous, we insert a [SKIP] placeholder between them for

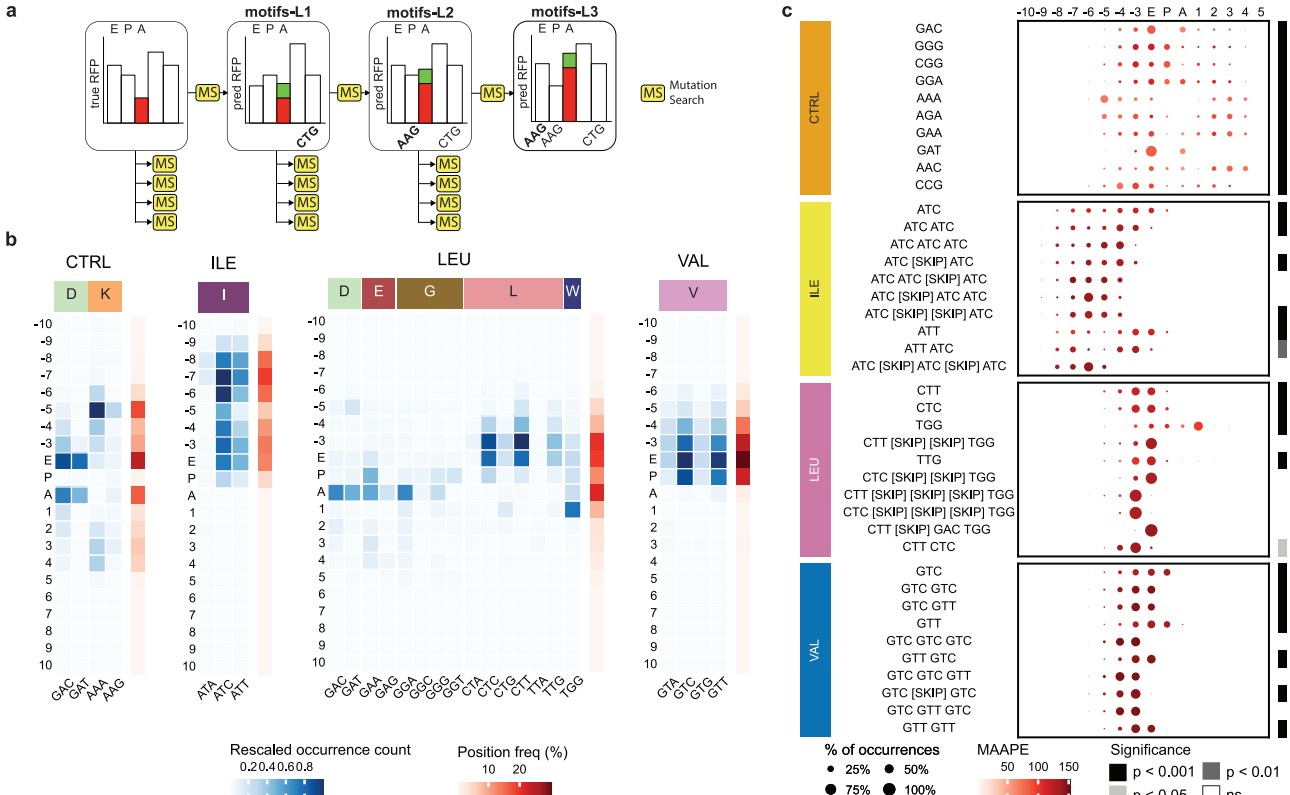

**Fig. 3 | Codon motifs extraction and analysis. a** Diagram of the beam search algorithm, explaining the process of extracting motifs using Riboclette. The algorithm performs a Mutation Search (MS) on all the codons in the Significance Window (10-codons on both directions from the A-site) for a given "non-peak" position - which is defined as a position in the gene where the RPC values are less than the gene RFP's mean minus one standard deviation. We conduct the MS five times to extract the top-5 mutations that increase the RPC value at the A-site. This process is repeated three times (referred to as the beam search width), with the extracted motifs being frozen before every new iteration. Using this pipeline, we extract motifs of length 1–3 codons that are referred to as "motifs-L1", "motifs-L2", and "motifs-L3", respectively. In subfigures **b, c**, we analyze motifs obtained by applying a beam search on 1000 genes for each condition. **b** Percentage of occurrence of the top codon motifs at each relative position to the A-site, ranked by the average Mean Arc-tangent Absolute Percentage Error (MAAPE). The occurrence count is normalized across the entire heatmap. The position frequency on the right reflects the sum of each row's occurrence count, normalized across all rows. **c** Frequency of codons in motifs at each position relative to the A-site. For each motif and position, the occurrence percentage is reported, normalized per motif. The average MAAPE is provided to indicate the relative increase in RPF counts when the given motif is added to the sequence. We also report the adjusted p-values from Fisher's exact test to assess the statistical significance of motif enrichment in the neighborhoods of RFP peaks compared to any other regions. A more comprehensive analysis, together with the remaining double and triple deprivations reported in Figs. S7–S12.

each codon position. For example, the VAL motif "GTC [SKIP] GTC" occurring at the −3 and E sites would be composed of two GTC codons separated by any other codon. Additionally, we validate these extracted motifs by conducting a Fisher's exact test to measure their enrichment near the peak regions as compared to any other regions in the RFP.

Studying the codon-wise distribution across motifs elucidates stalling codons in different deprivation conditions (Figs. 3B and S7–S9). For CTRL, Lys and Asp are the major contributors, with a bias for the AAA codon upstream, which characterizes the library preparation bias[36] noticed in the CTRL global attribution profile (Fig. 2A). Likewise, all the deprived conditions showcase the deprived codons to be the key determinants towards stalling, more clearly emerging for ILE and VAL than for LEU. Considering the milder nature of LEU deprivation, we observe residual effects from CTRL, including motifs of negatively charged amino acids (Asp and Glu) and Gly.

The identified motifs provide an additional layer of positional information, which can be studied to understand how interactions between different codons affect ribosome stalling (Figs. 3C and S10–S12). In the CTRL condition, we primarily observe motifs of length 1 coding for Asp, Glu, Arg, Lys, and Gly amino acids, which are also found to be significantly enriched in the RFP peak neighborhoods. This result corroborates our findings from the mean attribution analysis (Fig. 2C). Additionally, the

presence of the AAA codon at −5 can be ascribed to technical bias as discussed previously for the CTRL global attribution profile. For all the single amino acid deprivation conditions, the expected occurrence of deprived amino acids is integrated by the presence of longer motifs. Interestingly, as also seen in the analysis by ref. 17, motifs for LEU and ILE have higher long-range effects on stalling as compared to CTRL and VAL, with ILE motifs being found as far as codon position −9 from the A-site. The effect of distant, deprived LEU codons has been identified in other analyses[16]. This pronounced long-range effect of ILE codons could be attributed to their greater sparsity in the genome as compared to VAL codons (refer Fig. S13B). The LEU deprivation, which is the mildest, has Cys (C: TGC, and TGT) and Trp (W: TGG) codons near the A-site. This observation could potentially be explained by the fact that these three codons are near cognates of the stop codon TGA[37]. Double and triple deprivations follow a similar pattern as ILE and VAL, respectively (Fig. S12). Most motifs of length 1 and 2 could be validated through the statistical enrichment analysis ($p \leq 0.05$), whereas the majority of the length 3 motifs were not. This could mean that the beam search motif extraction pipeline is potentially picking up on more information than statistical methods, although experimental validation would be necessary to support this claim. The full processed results from the beam search motif extraction pipeline have been provided in Supplementary Data 1.

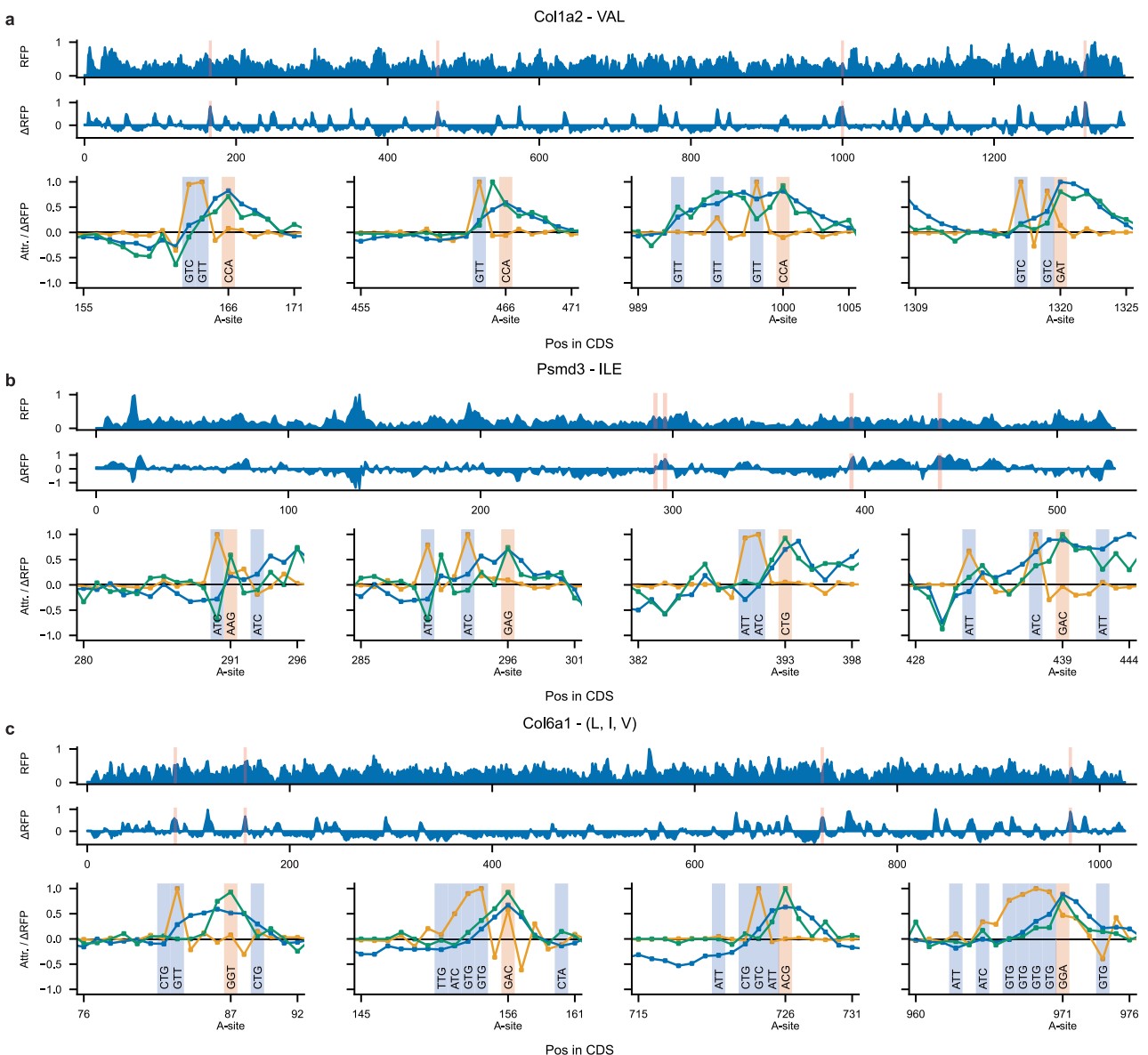

**Fig. 4 | Riboclette position-specific interpretability for Col1a2 (VAL), Psmd3 (ILE), Col6a1 (L, I, V).** This panel presents, from top to bottom, the predicted CTRL RFP, the predicted deprivation difference ΔRFP, and the attributions (orange), along with the true (green) and predicted (blue) ΔRFP around four selected A-sites. The attributions, together with the true and predicted ΔRFP, are max-normalized across the gene. The selected A-site is highlighted in red, while the deprived codons are highlighted in blue. **a** Col1a2 gene on Valine deprivation at positions 166, 466, 1000, and 1320. **b** Psmd3 gene on Isoleucine deprivation at positions 291, 296, 393, and 439. **c** Col6a1 gene on triple (L, I, V) deprivation at positions 87, 156, 726, and 971.

## Codon motifs help explain position-specific ribosome stalling

While current methods analyze stalling sites at a genome-wide or average level, Riboclette's codon attributions and motifs can be employed to analyze gene-specific peaks and uncover their determinants. We present a case study on three genes: *Col1a2* in VAL deprivation, *Psmd3* in ILE deprivation, and *Col6a1* in (L, I, V) deprivation (see Fig. 4). These genes were chosen from the differentially expressed genes in $\mathscr{D}^{int}$ (refer to "Data Processing" in the "Methods" section) as representative examples. On average, they show a significantly higher normalized read count on codons corresponding to the deprived amino acid compared to the CTRL condition. We investigate these three genes at four different RFP peak positions on the CDS by studying the true and predicted RFPs in relation to the Riboclette attributions to understand and explain their stalling patterns. We use the max-normalized true RFPs for conducting this analysis, as this allows direct comparison with the normalized predicted signal.

The attribution profiles highlight a strong influence from the deprivation codons present in the neighborhood of the perturbed A-site. We notice that GTC and GTT codons are primary influences in VAL and (L, I, V) deprivations, whereas ATT and ATC codons are highlighted in the ILE deprivation. As expected from our global attribution analysis (Fig. 2A), the codon attributions gradually reduce as the distance from the perturbed A-site increases. Although, as noticed previously, Riboclette is capable of taking deprivation-specific information into account where distant ILE codons are still capable of influencing the RPC at the A-site (Fig. 3B), as observed in the attribution profiles for the *Psmd3* gene at CDS positions 296 and 439. Another observation from the global attribution analysis, noticeable in different attribution profiles, is that the codons present upstream show a stronger influence on the A-site prediction compared to the ones present downstream. Riboclette is also able to factor in these lesser downstream codon attributions, which we

notice in *Psmd3* (CDS positions 29 and 439) and *Col6a1* (CDS positions 87, 156, and 971).

Importantly, we detect the presence of different motifs that we have extracted using the beam search pipeline in the attribution profiles, which could be used to explain the stalling at the RFP peaks. In the case of *Psmd3*, the ATC-[SKIP]-[SKIP]-ATC (CDS positions 291 and 296) and ATT-ATC (CDS pos 393) motifs are present. (L, I, V) deprivation motifs mirror that of VAL deprivation, so we observe common motifs such as GTT, GTG-GTG, GTC-[SKIP]-GTC, and GTT-[SKIP]-[SKIP]-GTT in the attribution profiles for *Col1a2* and *Col6a1*.

We notice an interesting phenomenon for ribosome stalling in the CDS position 971 for *Col6a1*, where the presence of multiple deprived codons present consecutively in the CDS causes a buildup of ribosomes, which causes a steady increase in RPC, with the greatest peak occurring after the motif subsequence. We term this the "ramp effect" for deprived codons, a pattern resembling previously reported linear additive effects for positively charged residues[38].

Riboclette, in most cases, can accurately identify and predict RFP peaks, but in some cases, these predictions might differ by a few positions, as observed in *Col1a2* CDS position 466 and *Psmd3* CDS position 393. Even in these situations, the attribution profiles still provide a plausible explanation for the stalling behavior. This case study outlines the utility and applicability of Riboclette attribution profiles, along with the identified codon and motif drivers to explain ribosome stalling patterns at a codon resolution in different deprivation conditions.

## Discussion

In this study, we present Riboclette, a framework for processing, predicting, and interpreting ribosome footprint profiles in different conditions. We applied Riboclette to understand translation elongation regulation in the context of essential branched-chain amino acids' deprivation, which is still an open area of research[17]. To the best of our knowledge, we are the first ones to conduct a large-scale machine learning based study to elucidate the ribosome stalling determinants in the context of multiple amino acid deprivations. Riboclette's novel double-head architecture enables the learning and prediction of multiple deprivation conditions with a single model, enhancing both predictive performance and interpretability. Since Riboclette predicts ribosome footprint profiles from input mRNA sequences, we leveraged it to impute missing values, further improving performance through pseudo-labeling by retraining the model on imputed data. Next, we used Integrated Gradients[26], a standard post-hoc feature attribution method, to examine the effect of the codon context around the ribosome A-site, highlighting that the codons in the protected fragment are the ones that mostly affect translation elongation. Moreover, aggregating the attributions codon-wise, highlighted codons associated with deprived amino acids, poly-basic regions, and negatively charged amino acids to be major contributors to ribosome stalling. Furthermore, using these attributions, we unravel how synonymous codon bias manifests in the three single deprivations. While attributions provide a global position and codon-wise perspective of translation elongation determinants, they do not account for interaction effects, that is, motifs. Inspired by counterfactual explanations which are used to explain the behavior of a given machine learning model by generating minimal input perturbations that change its output[34], we generated stalling-inducing in silico motifs from Riboclette. Hence, our extracted motifs identify both a set of up to 3 codons and their positions that impact translation elongation, providing a new precise tool to understand its determinants. Using these extracted motifs, we identify several interesting stalling related phenomenon such as the long-range stalling effects caused by LEU and ILE motifs, and the ramp effect for deprived codons, which explains how the occurrence of consecutive deprived codons leads to a greater stalling response.

Riboclette's embeddings contain valuable information regarding translation regulation and the effects of various cell stress conditions, providing a unique opportunity to investigate related mechanisms, such as protein co-translational folding and protein expression. These embeddings

could be used independently or in conjunction with ones from other genomic and protein language models[39] to generate further enriched translation-specific representations to better understand the downstream tasks. Furthermore, insights regarding stalling codons and motifs could be utilized to detect problematic codon subsequences and propose edits for optimizing mRNA sequences. This approach could also be extended to design a codon optimization framework aimed at increasing protein expression efficiency. Studying the effect of amino acid deprivations in the context of translation could help us understand more about cancer and chronic illnesses, where nutrient stress, including amino acid scarcity, have been shown to reshape translational programs and activate stress response pathways. In addition, we could study inherited metabolic disorders such as maple sirup urine disease, where a toxic accumulation of amino acids has been shown to disrupt protein synthesis. Furthermore, we can employ Riboclette to study the effect of deleterious mutations in the context of ribosome stalling and translation dysregulation. Such an analysis could potentially further our understanding regarding various translation-related metabolic disorders and help in designing targeted therapeutics[40].

Although Riboclette presents an accurate method to interpret the regulation of translation elongation in the context of amino acid deprivation, it has certain limitations. Riboclette can currently only predict and understand translation dynamics under six deprivation conditions. A more generalized model would be one that could extrapolate to new biological conditions, such as unseen amino acid deprivations, and stress conditions in different species and cell lines. Although a major challenge in developing such a model would be the scarcity of high-quality, large-scale pan-species deprivation-related Ribo-seq data for higher eukaryotes. Additionally, the Ribo-seq experimental pipeline embeds technical artifacts in the data (refer Fig. 2A) and merging different Ribo-seq datasets from different cell types can introduce batch effects (refer Fig. S4). The generalized model that we envision should also be capable of adapting to all these variations in the datasets to isolate and study only the biological signal. The current Riboclette framework was employed to learn more about translation elongation dynamics. Using the Riboclette attributions and the beam search pipeline, we extract various codons and motif-level drivers indicated in ribosome stalling. Although these drivers present a valuable approach to study ribosome stalling at codon resolution, all of these experiments have been conducted in silico. While statistical significance tests provide a layer of reliability to the identified motif drivers, in vitro validation is necessary for confirming motifs inferred by Riboclette that are not present in our dataset.

In summary, Riboclette, paired with the web server (see "Data Availability"), introduces a novel framework for condition and codon-wise understanding of translation elongation, leveraging its dual-head architecture, feature attributions, and motif extraction pipeline. We believe that our characterization of stalling at the codon level could further our understanding of translation elongation intragenic heterogeneity and its underlying determinants.

## Methods
### Experimental pipeline
We used two Ribo-seq datasets available on the Gene Expression Omnibus (GEO) database from *Mus Musculus*. The first dataset was collected from mouse liver under a chow diet ($n = 84$, GEO GSE73553)[41], and the second was obtained from NIH3T3 fibroblast cells (GEO GSE291653)[17]. The liver dataset was generated only in the control (CTRL) condition. The fibroblast dataset was generated under six conditions, which were CTRL, single deprivations of leucine (LEU, L), isoleucine (ILE, I), and valine (VAL, V), combined LEU, and ILE deprivation (dubbed as "double" deprivation, represented as (L, I)), and combined deprivation of all three of them (dubbed as "triple" deprivation, represented as (L, I, V)).

For the fibroblast dataset, SRA files were converted to FASTQ files, and adapter sequences were trimmed from the raw reads using cutadapt with parameter `-m` set to `10` and with the adapter sequence "TGGAAT TCTCGGGTGCCAAGG". Using an in-house Perl script, duplicated reads

carrying identical sequences and UMIs (four random nucleotides at both 5′ and 3′ ends) were removed. De-duplicated reads from the fibroblast dataset, along with raw data from the liver dataset, were pre-processed using the Ribo-DT snakemake pipeline[42], with mouse genome sequences (GRCm38/mm10) downloaded from ENSEMBL (Release 95), and read size between 26 and 35 nucleotides. Intermediate ".count" Ribo-DT output files were used to generate ribosome density profiles.

## Data pre-processing

We initiated our data processing pipeline by normalizing the Ribosome Profiling Counts (RPCs) across different replicates for each gene. Following this, we merged annotations from the replicates to generate the gene-wise Ribosome Footprint Profiles (RFPs). To ensure consistency, we retained only the longest transcript for each gene, resulting in a dataset where each gene is represented by a unique transcript. Subsequently, the RPCs underwent normalization against the average gene count and were $log(x + 1)$ transformed to reduce the RFPs skewness. To define the dataset used to train the machine learning models, we applied three filters. First, we excluded genes with sub-sequences of zero count values longer than 20 codons. Next, we removed sequences exceeding 2000 codons in length. Finally, we retained only those genes with coverage greater than 30%, where coverage is defined as the ratio of annotated (non-NaNs) codons with non-zero values to the total number of annotated codons.

The resulting dataset consisted of 22149 samples, with 5285 distinct genes, where samples here refer to a gene in a specific condition. This dataset was then split into training, validation, and testing sets, ensuring that the last two contained only well-annotated genes. This was conducted by initially sorting the genes in the dataset in descending order of their coverage. Following which, we started at the top of this list and assigned alternating genes sequentially to the training, validation, and testing sets. This process continued until the validation and testing sets contained 5% and 20% of the original dataset, respectively. The training, validation, and testing sets are denoted as $\mathscr{D}^{train}$, $\mathscr{D}^{val}$, and $\mathscr{D}^{test}$.

To perform the post-hoc interpretability studies, we extended our dataset with an extra set of genes (represented as $\mathscr{D}^{int}$). This set consists of 1734 samples which are chosen based on two conditions. The first set of 145 samples were those that were discarded in the dataset processing pipeline because they were longer than 2000 codons in length, and the second set of 1589 samples were identified by conducting a differential expression analysis. Briefly, genes were selected by performing likelihood ratio tests between two models. The null model assumes that all codons respond similarly to the deprivation, whereas the alternative model allows the codons corresponding to the deprived amino acid to have a different mean. The likelihood ratio statistic, which follows a chi-square distribution, was calculated using the normalized RFPs. Statistics regarding coverage and sequence length distributions for $\mathscr{D}^{train}$, $\mathscr{D}^{val}$, $\mathscr{D}^{test}$, and $\mathscr{D}^{int}$ can be found in Fig. S1.

## Riboclette

Riboclette takes as input the mRNA sequence $s = (c_1, ..., c_n)$, composed by $n$ codons $c_i$, and a deprivation condition identifier $d$ corresponding to one of the conditions in the training set. The codons and the condition identifier are one-hot encoded and passed through separate learnable embeddings of the same size, which is optimized as a hyperparameter. For the model backbone, we have implemented the Bidirectional LSTM (BiLSTM)[43,44] (Riboclette BiLSTM) and the XLNet[45] (Riboclette Tr).

The XLNet applies multiple Relative Multi-Head Attention (RMHA) blocks, employing the implementation from HuggingFace's transformer library[46] of the Transformer-XL model[47], which first introduced the Relative Positional Encodings (RPE). More in detail, given the input of an attention block $x$, the XLNet's output is $h = (h_1, ..., h_N)$, where $h_i = \sum_{j=1}^{n} \alpha_{ij}(x_j W^V)$, with $\alpha_{ij} = \exp(e_{ij})/\sum_{k=0}^{N} \exp(e_{ik})$, where $n$ are the input tokens and $W^V \in \mathbb{R}^{D \times D}$. Following the definition of RPE's in Transformer-XL: $e_{ij} = (x_i W^Q + u)(x_j W_E^K)^T + (x_i W^Q + v)(r_{i-j} W_R^K)^T$, where $r_{i-j} \in \mathbb{R}^D$ is

the relative positional encoding, $u, v \in \mathbb{R}^D$ are trainable parameters, and $W^Q, W_E^K, W_R^K \in \mathbb{R}^{D \times D}$ are weight matrices.

The samples in the dataset are defined with an input pair consisting of the mRNA sequence and amino acid deprivation condition $(s, d)$, and a pair consisting of the RFPs in the control (CTRL) and deprived (DC) conditions $(y_{CTRL}, y_{DC})$. When the input sample is the control condition ($d$ is control) $y_{CTRL} = y_{DC}$. From the pair of RFPs, we introduce the pair of target labels $(y_{CTRL}, y_{\Delta D})$, where $y_{\Delta D} = y_{DC} - y_{CTRL}$. Riboclette's CTRL and $\Delta D$ heads predict the target labels as $(\hat{y}_{CTRL}, \hat{y}_{\Delta D})$.

## Pseudo-labeling

Some samples in the training set have low-coverage RFPs, and this might negatively impact Riboclette's training stability. To address this issue, we applied pseudo-labeling[27]: initially, a teacher model was trained to impute the missing RFPs, followed by training a student model on the imputed dataset. As a teacher model, we used an ensemble of 5 Riboclette Tr initialized with different seeds. Given the prediction $\hat{y}_{DC}^s$ of the model with seed $s \in S$, we define the ensembled prediction as $\hat{y}_{DC}^* = \sum_{s \in S} \hat{y}_{DC}^s / |S|$. Then, for each sample $y_{DC}$ in the training set, we imputed the missing RPCs using the predicted count in $\hat{y}_{DC}^*$, that is $y_{DC}(i) := \hat{y}_{DC}^*(i)$ if $y_{DC}^i(i) = \text{NaN}$, where $y_{DC}(i)$ is the RPC of the $i$-th codon. We refer to the student model re-trained from the imputed training dataset as Riboclette Tr-Pl.

## Baselines

In our experiments, we tested three versions of Riboclette (BiLSTM, Tr, Tr-Pl). To benchmark against existing machine learning methods to predict RFPs, we selected RiboMIMO[24], Riboformer[23], and Iχnos[28]. Note that since these three models are not designed to predict multiple conditions together, they are trained on one condition at a time. To train these existing baseline models, we used the suggested default values for the hyperparameters. For Riboformer, we implemented a sequence-only version of it without the starting condition RFP. Given that both Riboformer and Iχnos truncate codons from the beginning and end of the sequence during pre-processing, for the performance comparisons, we standardized pre-processing across methods by applying the more extensive Riboformer transformation, removing the first and last 20 codons, to all methods.

## Riboclette hyperparameter tuning

Hyperparameter optimization for each Riboclette variant was performed through the Optuna library[48]. The tested configurations are outlined in Table S2. Optuna uses results from previous trials in combination with Bayesian optimization techniques to determine the most optimal set of hyperparameters for the next set of trials, thus reducing the number of hyperparameter combinations that need to be tested. In order to speed up this process, we parallelized the trials with 10 parallel Optuna agents that traversed the hyperparameter search space concurrently.

## Model Training and Evaluation

Riboclette BiLSTM and Riboclette Tr are trained with a loss (Eq. (1)) consisting of the Mean Absolute Error (MAE) applied to the deprivation RFP prediction and a Pearson Correlation Coefficient (PCC, $r$) based loss applied to all three of the CTRL, $\Delta D$, and deprivation RPC predictions. All of these models are primarily evaluated based on the Pearson correlation coefficient between the predicted and the ground truth deprivation RFP (Eq. (2)).

$$\mathscr{L} = 3 - r(y_{CTRL}, \hat{y}_{CTRL}) - r(y_{\Delta D}, \hat{y}_{\Delta D}) - r(y_{DC}, \hat{y}_{DC}) + |y_{DC} - \hat{y}_{DC}| \tag{1}$$

$$r(y, \hat{y}) = \frac{\sum_{i=1}^{n} (y_i - \bar{y})(\hat{y}_i - \bar{\hat{y}})}{\sqrt{\sum_{i=1}^{n} (y_i - \bar{y})^2} \sqrt{\sum_{i=1}^{n} (\hat{y}_i - \bar{\hat{y}})^2}} \tag{2}$$

In addition to the PCC, we also employed Mean Arc-tangent Absolute Percentage Error (MAAPE)[35] (Eq. (3)), Mean Squared Error (MSE) (Eq.

(4)), and Spearman Correlation Coefficient (SCC, $\rho$) (Eq. (5)) for model evaluation. The $R$ in Eq. (5) represents the rank operator.

$$MAAPE(y, \hat{y}) = \frac{100}{N} \sum \arctan\left(\frac{|y - \hat{y}|}{y}\right) \tag{3}$$

$$MSE(y, \hat{y}) = \frac{1}{n} \sum_{i=1}^{n} (y_i - \hat{y}_i)^2 \tag{4}$$

$$\rho(y, \hat{y}) = 1 - \frac{6 \sum_{i=1}^{n} d_i^2}{n(n^2 - 1)}, \quad d_i = R(y_i) - R(\hat{y}_i) \tag{5}$$

### Explaining Riboclette predictions with codon attribution scores

The Riboclette model, with its two output heads, provides us with a framework that makes it easy to visualize, interpret, and study the effects of different deprivation conditions at a single-codon resolution. In order to understand what the model takes into account to make its prediction, we use the Integrated Gradients[26] algorithm implemented in the Captum library[49] to extract these attribution vectors. For each gene and each position $i$ in the CDS, we extract two separate attribution vectors, one each from the predictions of the CTRL and $\Delta D$ heads, they are referred to as $\mathscr{G}_{CTRL}^i$, $\mathscr{G}_{\Delta D}^i$ respectively. The Integrated Gradients algorithm, given a gene sequence of length $l$ codons and a selected position $i$, provides as output an attribution vector of shape [$l$, d_model], with d_model being the codon embedding size. The codon embeddings are then summed up to obtain a vector of codon-wise attributions of shape [$l$, 1]. $\mathscr{G}_{CTRL}^i[j]$, and $\mathscr{G}_{\Delta D}^i[j]$ are the CTRL and $\Delta D$ attribution vectors representing the attribution assigned to the $j$th codon in predicting the RPC of the $i$th codon.

### Extracting codon motifs as Riboclette counterfactual explanations

The Riboclette model was used to conduct codon mutations in order to extract motifs causing ribosome stalling. This procedure is analogous to extracting counterfactual explanations[34] for interpreting machine learning models, where the goal is to identify the smallest possible change to the input that alters the model's output. In our scenario, the aim is to find codon mutations that generate a strong increase in the predicted RPC in a given position.

Mutations are performed on 6000 fast gene windows selected from the full dataset, randomly selecting 1000 windows from each of the six conditions. These windows are centered on the A-site, including 10 codons upstream and 10 codons downstream, with a total length of 21 codons. Specifically, a window is classified as "fast" if the RPC at the A-site was lower than the gene's RFP average minus one standard deviation. Similar thresholds have been used in previous studies to distinguish fast and slow positions[24,50].

To extract the motifs of interest from the chosen gene windows, we conduct a mutation beam search of width 5, with the maximum mutation size being set to 3. These parameters dictate that the search first finds for each of the windows, 5 single codon mutations that result in the greatest increase in RPC at the A-site, these are referred to as "motifs-L1". Since the initial counts could be close to zero, we overcame this issue by using MAAPE[35] to measure the RPC increase. In the second step, for each of the single mutation windows in motifs-L1 by freezing the already mutated codon, we find 5 more single mutations that result in the greatest increase in ribosome counts at the A-site. The resulting 25 mutation pairs are referred to as "motifs-L2". In the final step, for each pair mutation windows in motifs-L2 by freezing the already mutated codon pair, we find 5 more single mutations that result in the greatest increase in ribosome counts at the A-site. The resulting 125 mutation triples are referred to as "motifs-L3". Finally, taking the mutations of all lengths, we produce a total of 155 motifs for each of the 6000 windows. The A-site RPCs for the mutated gene windows in this beam search are predicted using Riboclette's CTRL head for those genes in the control condition and Riboclette's $\Delta D$ head for those genes where there is an amino acid deprivation.

Motifs spanning multiple codons may include non-contiguous codon positions. To account for these gaps, we insert "[SKIP]" tokens between mutated codons when defining a specific motif. For example, if our algorithm identifies a motif with "CTG" at position −1 and "CTT" at position 1, we represent it as "CTG [SKIP] CTT" at position −1. Note that for motifs, positions are relative to the A-site, e.g., −1 corresponds to the P-site.

To assess whether the motifs identified by Riboclette are enriched near RFP peaks in the dataset, we analyzed each condition by counting how often each motif appeared within the Significance Window of a peak (defined as 10 codons upstream and downstream of the A-site) versus outside of it. Using the total number of codon positions inside and outside the Significance Windows, we constructed a contingency table for each of the 50 most frequent motifs. We then applied a two-sided Fisher's exact test to evaluate enrichment, controlling for the False Discovery Rate using the Benjamini–Hochberg correction.

### Statistics and reproducibility

There are two ribo-seq datasets used in this study, which are from the liver and fibroblast cells. The liver dataset consists of 84 replicates, and each different condition in the fibroblast dataset consists of 3 replicates. For the statistical testing to validate the presence of Riboclette identified motifs, we used the Fisher's exact test, and the p value correction was conducted using the Benjamini–Hochberg method.

### Reporting summary

Further information on research design is available in the Nature Portfolio Reporting Summary linked to this article.

## Data availability

All the data is publicly accessible through the Gene Expression Omnibus (GEO) database at https://www.ncbi.nlm.nih.gov/geo with the following accession numbers: GEO [GSE73553[41], GSE291653[17]]. The preprocessed data, model checkpoints, and attributions have been made available on Zenodo[51]. The online server to visualize Riboclette predictions and attributions can be accessed at https://lts2.epfl.ch/ribotly.

## Code availability

The code to replicate the results, the weights for the pre-trained models, and the processed dataset can be found at https://github.com/fcraighero/riboclette.

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

## Acknowledgements

This work was supported by a Swiss National Science Foundation Sinergia grant (CRSII5-205884). Moreover, we thank Ali Hariri, Maria Boulougouri,

Anaís Haget, Nicolas Aspert, and David Neill Asanza for their valuable insights and discussions regarding the project, and in particular Nicolas Aspert for his support in building the web server.

## Author contributions

C.G., M.V.N. and F.C. designed the study. L.W. conducted the ribo-seq experimental assays to generate the amino-acid deprivation related dataset. C.G. processed the initial experimental data. M.V.N. and F.C. developed the Riboclette machine learning model and conducted the model training, benchmarking with the state-of-the-art models, and interpretability analysis. M.V.N. and F.C. wrote the manuscript, and C.G., F.N. and P.V. edited it. C.G., F.N. and P.V. supervised the study. All authors contributed to the subsequent revisions.

## Competing interests

The authors declare no competing interests.
