## [Transparent Peer Review File · Communications Biology]

Conditional Deep Learning Model Reveals Translation Elongation Determinants during Amino Acid Deprivation

Corresponding Author: Mr Mohan Vamsi Nallapareddy

Version 0:

Reviewer comments:

Reviewer #2

(Remarks to the Author)

Reviewer #1 and I shared similar concerns regarding the model's generalizability and the lack of motif validation. The authors have provided computational validation for their motif analysis. As for the model's limited scope and generalizability, the authors offered a detailed explanation, attributing it to the scarcity of datasets under diverse amino acid deprivation conditions. While this issue has not been fully addressed, I think the authors' explanation is reasonable, and the model itself may offer significant value to the field.

The authors have also addressed my other concerns.

I recommend this manuscript for publication in Communications Biology.

Reviewer #3

(Remarks to the Author)

1. Although the author provided point-by-point responses to my questions, the revisions to the content are limited. In particular, the performance comparison with existing methods remains insufficient.

2. For my concern on sufficient performance metric, the author supplement only one additional performance metric—Mean Arctangent Absolute Percentage Error (MAAPE), in Fig. A2. I still suggest to include other more evaluation metrics.

3. The revised content were not highlighted, making it difficult to identify which parts have been modified by the author.

Version 1:

Reviewer comments:

Reviewer #3

(Remarks to the Author)

I think all my concerns have been addressed.

We thank the reviewers for providing detailed comments and crucial feedback on our manuscript. The reviewers expressed their positive feedback on the development and performance of the proposed Riboclette method, while also raising concerns regarding its novelty, generalizability, and the lack of validation concerning the motif analysis. Down below, we address all the reviewers' comments individually:

Reviewers' Comments:

Reviewer #1 (Remarks to the Author):

This paper presents Riboclette, a new conditional deep learning model designed to predict ribosome footprint profiles (RFPs) from mRNA coding sequences under various amino acid deprivation conditions. The model features a dual-output architecture that separately models baseline (control) and deprivation-specific translation elongation dynamics. By leveraging integrated gradients and in silico motif extraction, the authors identify codon- and motif-level determinants of ribosome stalling. Riboclette outperforms prior models in predictive accuracy and interpretability, and is accompanied by an interactive web platform for exploration and analysis. The method has some novelty. It also makes effective use of interpretability techniques, including integrated gradients and beam search for motif discovery. The authors benchmark Riboclette against existing models (e.g., RiboMIMO), demonstrate performance across multiple deprivation conditions, and highlight practical applications through gene-specific case studies. The release of open-source code and a web-based visualization platform enhances the reproducibility, transparency, and accessibility of the work.

Response: We thank the reviewer for their comprehensive analysis of the manuscript and for their positive feedback regarding the performance of the proposed Riboclette model and the accessibility of the work.

However, a major limitation of the study is the narrow scope and limited generalizability of the model. Riboclette is trained and evaluated on just six predefined amino acid deprivation conditions, all derived from mouse fibroblast and liver datasets that are already publicly available. This raises questions about the practical utility of the model, particularly since the conditions being modeled are already directly observable using the same Ribo-seq data that the model is trained on. Traditional statistical or mechanistic analyses could yield similar insights without the complexity of a deep learning model.

Moreover, Riboclette's ability to generalize to unseen deprivation levels, alternative amino acid combinations, different stress conditions, or other species is not demonstrated and likely limited. Because deprivation conditions are represented categorically, the model lacks the flexibility to interpolate or extrapolate to new biological settings; for instance, predicting the effect of partial leucine deprivation or starvation of an unmodeled amino acid. This undermines the claim that Riboclette serves as a broadly applicable tool for studying translation elongation under stress.

Response: We appreciate the reviewer's suggestion to demonstrate Riboclette's generalizability potential by extrapolating it to new biological conditions (stress conditions, and unseen amino acid deprivations). However, we believe that this would be quite challenging to validate, considering the lack of large-scale, high-quality amino acid

deprivation related ribosome profiling datasets. The dataset we have studied in this manuscript is the largest, with 5 branched-chain amino acids (BCAAs) deprivation conditions and a control. This dataset was made publicly available together with the current manuscript through a parallel study [Worpenberg et. al., [10.1101/2025.03.27.645654](https://doi.org/10.1101/2025.03.27.645654)]. Additionally, this is the only ribosome profiling dataset that studies the effects of combined BCAA deprivations. To the best of our knowledge, we are the first to conduct a machine learning based study on discovering the determinants of ribosome stalling under amino acid deprivation conditions. The proposed Riboclette model could be utilized to predict RFPs and extract determinants of ribosome stalling for any gene in the mouse genome in these six different conditions. This can be conducted with ease using the publicly available Riboclette server or the ready-to-use Python PIP package. The model could also be applied to design variants of these mouse genes that could translate more efficiently, or to study metabolic disorders in mice that affect translation.

In order to conduct these generalizability studies, we have looked into other publicly available datasets that captured the effects of amino acid deprivations, for example, the one provided by [Darnell et. al., [10.1016/j.molcel.2018.06.041](https://doi.org/10.1016/j.molcel.2018.06.041)]. This dataset studies human cells in 2 amino acid deprivations (Leucine and Arginine), and a control. After analyzing this dataset, we deemed it unusable for our study, as we noticed that the leucine deprivation signal was weak, and the dataset sequencing depth was quite low.

We agree with the reviewer that generalizability is key to understanding the practical utility of a model; hence, we have added more details regarding this in the discussion section (**Section 3**) of the manuscript.

A further limitation concerns the motif extraction approach based on counterfactual perturbations. While the strategy of identifying codon substitutions that increase predicted ribosome occupancy is interesting, its biological relevance is unclear. Because the method relies entirely on the model's internal representations, it remains uncertain whether the identified motifs actually cause ribosome stalling in vivo, or simply reflect patterns the model has learned from the data. The paper does not rigorously validate these motifs against experimental ground truth. Given that ribosome profiling already provides high-resolution data on stalling positions, it would be appropriate to compare the model-derived motifs with sequence patterns enriched at empirically observed RFP peaks. Without such comparison, it is difficult to assess whether these motifs represent meaningful biological determinants or are artifacts of the model's training data and inductive biases.

Response: We agree with the reviewer's comment regarding validating the identified codon motifs, as it would make the findings from the model more reliable. Considering this, we validated the statistical significance of each motif's enrichment in the neighborhood of RFP peaks (protocol described in **Section 4.9** and results displayed in **Fig. 3c**). In detail, we counted how often each motif appeared within the Significance Window (10 codons) surrounding a peak, compared to its occurrences elsewhere. Using these counts, we constructed a contingency table and applied Fisher's exact test to each of the top 50 motifs per condition, with adjustments for false discovery rate. The significance of the statistical test was added to all the motif heatmaps (**Fig. 3c, Supp Fig A7-9**), while the contingency tables, odds ratios, and p-values are reported in the newly added **Supplementary Table A3**. We find that most of the motifs of length 1 and 2 are found to be statistically significant, as

opposed to motifs of length 3. The results from the statistical significance test provide key reliability metrics for the identified motifs. The motifs of length 3, which could not be validated through the statistical testing, could indicate novel findings from our beam search pipeline, although experimental validation would be required to verify this claim. We agree with the reviewer that experimentally validating these motifs would greatly benefit this analysis, but we believe that this would be out of scope for the current manuscript.

Reviewer #2 (Remarks to the Author):

In this study, the authors presented a deep learning model, namely Riboclette, for predicting and interpreting ribosome profiling signatures across amino acid deprivation conditions. This model proposed a double-headed architecture, with one head dedicated to control conditions and the other to deprivation-induced delta predictions, respectively. Riboclette integrated ribosome profiling prediction and imputation during the training process to promote the predictive performance. Additionally, the authors explored two model interpretation methods, including integrated gradients-based attribution and mutation-based motif search, which were used for understanding the underlying sequence/codon determinants of ribosome stalling.

Overall, Riboclette represents a novel computational framework capable of encoding, predicting, and interpreting ribosome profiling data under both control and amino acid deprivation conditions.

Response: We thank the reviewer for their detailed comments regarding the manuscript, and we appreciate the positive feedback regarding the novel design of the model framework for predicting and interpreting RFPs in different amino acid deprivation conditions.

However, there are several limitations of the current study, including the conceptual novelty of its research goal, the generalizability of the model, the lack of experimental validation, and the biological/disease significance or utility of its findings. My specific comments are as follows.

Major:

1. The model aims to predict and interpret ribosome stalling signals between control and amino acid deprivation conditions. However, to me, the model does not appear to offer a significant advance over Riboformer, which is also capable of modeling ribosome profiling between one condition versus another (can also be used for the setting of deprivation conditions) based on their differences. Likewise, Riboformer also explored the sequence determinants of ribosome pause/stalling events. Therefore, it is important to specifically point out and demonstrate the conceptual novelty or difference between Riboclette and Riboformer. Moreover, it is necessary to compare these two models, particularly since both models studied the sequence determinants of ribosome pausing/stalling.

Response: We understand the reviewer's concerns regarding the differences with the already existing Riboformer framework. We believe that there are several differences and improvements to Riboclette as compared to Riboformer, which are highlighted below:

- Riboformer predicts the target condition RPC using both the mRNA sequence window and the starting condition RPC. Whereas, Riboclette requires only the mRNA sequence. This is a key difference that limits a fair performance comparison. While

Riboformer mentions a sequence only variant [DOI: <https://doi.org/10.1038/s41467-024-46241-8>: **Supplementary Table 4**], to the best of our knowledge, the code for this is not publicly available.

- Riboformer bases its predictions and interpretability only on an mRNA window of 40 codons, whereas Riboclette studies the entire mRNA sequence. This allows us to conduct sequence-wise analyses, obtaining the whole ribosome density profile in one forward pass (**Fig. 4**).
- We propose a more robust motif extraction inspired by counterfactual explanations. The key differences in the motif extraction pipeline are described below:
 - Our beam search pipeline extracts motifs with minimal perturbations that cause the maximal increase in the ribosome stalling response around an RFP non-peak A-site position. This enables us to make minimal modifications to a target mRNA, lowering the risk of encountering out-of-distribution sequences, where the model's predictions are likely to be highly uncertain and prone to errors. Whereas the Riboformer approach conducts an enrichment analysis based on randomly perturbing a 10-codon window, potentially leading to sequences quite different from the training data, where the model has less confidence.
 - Our pipeline identifies many potential motif candidates, each characterized by a set of codons at a certain set of positions. Conversely, Riboformer clusters the 10-codon window perturbations with K-means, then extracts the amino acids that are mostly enriched in each of the K clusters. We argue that such an approach, compared to ours, investigates up to K “global effects” and does not explicitly capture codon interactions, while we identify precise combinations of codons and their positions for each of the deprived conditions. As a result, we might be able to capture complex effects, such as “CTC [SKIP] [SKIP] TGG” in Leucine deprivation (**Fig. 3C**).

We agree with the reviewer that it is necessary to discuss relevant work in more detail in the manuscript, and we have added additional details in **Section 2.1** to include this.

2. The investigation and modeling of amino acid deprivation were based on only one specific dataset obtained on NIH3T3 cells, making it hard to evaluate its robustness and generalizability. Therefore, the biological contexts in which deprivation occurs, such as across different tissues or cell lines, need to be expanded. This would be important to have since the deprivation effects on translation elongation can vary across contexts, given the different gene expression and hence mRNA pools (ie. codon needs) across contexts.

Response: We appreciate the reviewer’s suggestion to demonstrate Riboclette’s generalizability potential by extrapolating it to new biological contexts (different species and cell lines). However, as discussed before, we believe that this would be quite challenging to conduct, considering the lack of large-scale, high-quality pan-species amino acid deprivation-related ribosome profiling datasets. To the best of our knowledge, we conducted the largest machine learning based study to detect ribosome stalling determinants in the context of 5 different amino acid deprivation conditions. As mentioned previously, we have looked into other similar datasets studying amino acid deprivation, but we didn’t employ them in our study because they were of low quality, either in terms of the strength of the deprivation or the sequencing depth of the experiment.

3. The key observation and conclusion drawn from the study seem to be largely as expected. I wouldn't be surprised that deprivation of an amino acid would induce the ribosome stalling at the corresponding codons, or that the sequence/motif near a ribosome stalling site would have a larger contribution. While the modeling framework performs competently, the authors need to point out the novel findings and new insights that their model and analyses uncovered.

Response: We thank the reviewer for their positive feedback regarding the performance of the proposed Riboclette model. Although we understand the reviewer's comment regarding the novelty of the study's findings, we believe that we have gained several interesting new insights through this study, which we highlight below:

- We notice an interesting synonymous codon bias where we identify that only specific subsets of deprived codons drive the majority of the stalling response (**Fig. 2C.**).
- We are the first to study combined amino acid deprivations through the lens of machine learning, and we notice that these combined deprivations follow a similar pattern to that of the individual amino acid with the strongest stalling response. VAL has the strongest stalling response, followed by ILE, and then LEU (**Fig. 2B., and A6.**). Owing to this, the triple deprivation (L, I, V) mirrors VAL, and double deprivation (L, I) mirrors ILE.
- Through the motifs extracted from the beam search pipeline, we notice that motifs for some deprivations, such as LEU and ILE, display more long range effects on stalling as compared to the ones in the CTRL, and VAL (**Fig. 3B.**). We attribute this to the higher distance between the codon occurrences in the genome for ILE and LEU as opposed to VAL (**Fig. A13B.**).
- We identify an interesting phenomenon, which we term as the "ramp effect for deprived codons", with the additive increase in stalling induced at the A-site caused by the occurrence of consecutive deprived codons (Described in **Section 2.5** and mentioned in **Fig. 4C. pos 971**).

We have now edited the discussion (**Section 3**) to further highlight these novel findings from the study as suggested.

4. The biological importance or utility of this model is not very clear to me. In what biological or disease contexts would amino acid deprivation take place? How would the research of the sequence determinant of ribosome stalling in such deprivation conditions deepen our understanding of disease etiology or help with therapeutics? These questions need further clarification and ideally demonstration by examples.

Response: We thank the reviewer for their valuable comment on the importance of clearly mentioning the biological utility of the proposed model. In this study, we focus on elucidating the determinants of translation elongation rates, which remain complex and incompletely understood. However, it is increasingly clear that elongation, particularly through mechanisms such as ribosome stalling and collision, plays a central role in mRNA surveillance and ribosome-associated quality control (RQC), making system perturbations a powerful strategy to investigate how context modulates ribosome elongation dynamics. In addition, amino acid availability is a critical determinant of cellular homeostasis, and its

deprivation has profound biological consequences. In cancer and chronic illnesses, nutrient stress, including amino acid scarcity, can reshape translational programs and activate stress response pathways. Inherited metabolic disorders such as maple syrup urine disease exemplify how imbalanced amino acid metabolism leads to toxic accumulation and disrupted protein synthesis. Moreover, dietary interventions and experimental therapies that modulate amino acid levels are increasingly explored for their potential to target tumor metabolism or promote healthy aging, highlighting the translational relevance of understanding how amino acid deprivation influences translation elongation. We have now incorporated these considerations into the discussion (**Section 3**), highlighting both the complexity of elongation rate regulation and the relevance of biological contexts such as amino acid deprivation and disease-related perturbations.

5. In Fig. 2c, the authors found that "The combined deprivations mirror the amino acid leading to the strongest stalling". I was wondering if this is a general rule across different biological contexts or specific to the cell line that the author built the model on? Again, expanding the biological contexts (eg. different cell lines) of deprivation would be important to answer this question. Moreover, it is not immediately understandable why VAL codons (GTA/GTC/GTG/GTT) show stronger effects in ILE or LEU deprivation conditions compared to CTRL? Is there any general contribution of VAL codons on amino acid deprivation-induced stalling?

Response: We thank the reviewer for their valuable comment on understanding the mechanism of the combined deprivations. We notice that VAL codons have the highest mean RPF counts, followed by ILE and LEU codons (**Fig. 2B.**). This finding can be used to characterize the magnitude of the stalling response in these deprivations. Hence, we believe that the stalling response of the combined deprivations is directed by the constituent amino acid that causes the strongest individual stalling response. We have studied amino acid deprivations only in mice fibroblast cells, and in order to make a more general claim regarding this, we would have to study the effect of amino acid deprivation on translation in different biological contexts. Although, as mentioned before, this would be quite difficult to achieve considering the lack of high-quality datasets of this nature. The other datasets studying amino acid deprivation were not employed in our study because they were of low quality, either in terms of the deprivation signal strength or the sequencing depth of the experiment.

6. The beam-search-based strategy for identifying motifs that contribute to ribosome stalling sounds reasonable, but it is still important to validate the identified motifs as well as the distant regulation effect of upstream codons (eg. in ILE context) on stalling.

Response: We thank the reviewer for their suggestion to validate the identified motifs to help make the findings more reliable. We improved the current version of our manuscript by validating the statistical significance of each motif's enrichment in the neighborhood of RFP peaks (protocol described in **Section 4.9** and results displayed in **Fig. 3C.**). In detail, we counted how often each motif appeared within the Significance Window (10 codons up- and downstream) surrounding a peak, compared to its occurrences elsewhere. Using these counts, we constructed a contingency table and applied Fisher's exact test to each of the top 50 motifs per condition, with adjustments for false discovery rate. The significance of the statistical test was added to all the motifs heatmaps (**Fig. 3C., Supp Fig. A7-9**), while the

contingency tables, odds ratios, and p-values are reported in the newly added **Supplementary Table A3**. We find that most of the motifs of length 1 and 2 are found to be statistically significant, as opposed to motifs of length 3. The results from the statistical significance test provide key reliability metrics for the identified motifs. The motifs of length 3, which could not be validated through the statistical testing, could indicate novel findings from the beam search pipeline, although experimental validation would be required to verify this claim. Regarding the long-range effect of ILE codons, we already discussed in the first version of the manuscript a possible justification related to the higher sparsity of ILE codons (see **Fig. A13b**). Additionally, certain long motifs, such as “ATC [SKIP] [SKIP] [SKIP] [SKIP] ATC”, emerge as statistically significant in the updated analysis, exhibiting an odds ratio greater than 7 and a p-value below 0.001 (see **Table A3**).

Minor:

1. In Fig. 2A, the authors showed the attribution of the codon distance from A-site for the position with ribosome stalling signals. It'd be interesting to do a comparison with the positions without stalling.

Response: We thank the reviewer for the suggestion. We included the requested comparison in **Rebuttal Figure 1**. As this analysis considers the top-5 positions where the model is looking to predict the RPC (Δ RPC), we can see that the ranking and frequency are independent of the expected RPC (Δ RPC) values. This can be intuitively explained by considering that the model targets specific codon positions to identify stalling codons (see, e.g., the motif analysis in **Fig. 3B**). Based on the presence of these codons, the model then estimates the RPC (Δ RPC).

Rebuttal Figure 1. Comparison of A-site attributions around peak and non-peak codon positions. Figure comparing the results from **Fig. 2A**, showing the distribution of the top-5 most salient positions around peaks (top row) and fast regions (bottom row), for both the CTRL and ΔD head. Peaks are identified by considering the codon positions where the RPC (Δ RPC) is greater than the gene RFP's mean plus one standard deviation. Conversely, non-peak codon positions are the ones where the RPC (Δ RPC) is less than the gene RFP's mean minus one standard deviation. The code to reproduce the figure has been added to the code repository.

2. Fig 3C was cited before Fig 3B in the manuscript. This needs to be fixed.

Response: We thank the reviewer for identifying this. We have now edited the text to cite and discuss **Fig. 3B**, before **Fig. 3C**.

3. For data accessibility, GSE291652 and GSE291653 are not available.

Response: We thank the reviewer for checking the data availability. Having open-source code and data is pivotal to our study. We only use GSE291653, and we had made a mistake by mentioning GSE291652. We have edited the manuscript to reflect this. The mentioned data is in private mode in the GEO database. We provide a private token ("*cnwtoaqalbotncd*") for the reviewers to access it.

4. The label of the y-axis of Fig. 2b is missing. Also, should the x-axis of middle and right panels be "Mean RPF Counts" or "Delta RPF"?

Response: We thank the reviewer for identifying this, and we have edited **Fig. 2B**, to add the missing legends for the axes.

5. The authors mentioned that "the Significance Window was normalized and examined" for Fig. 2, but the specific method for how the normalization was performed is not provided in the Methods.

Response: We thank the reviewer for suggesting that more information about the normalization protocol be provided. The attribution profiles of the codons in the significance windows were "max normalized" and then studied for the interpretability analysis in these figures. This clarification has been added to the **Fig. 2** caption: "*These peaks were considered to be the A-sites, and the surrounding attributions in a 10-codon window up- and downstream (the Significance Window) were max normalized and examined*".

6. The annotation 'RNA expression expr' in Table A1 should be revised to 'RNA expression (expr)' for proper formatting.

Response: We thank the reviewer for identifying this, and we have edited the caption of Table A1 accordingly.

Reviewer #2 (Remarks on code availability):

The code is well organized and annotated.

The web server implementation is plausible. The authors may further clarify whether the framework supports user-uploaded Ribo-seq data for tailored tasks, specify hardware requirements for local deployment, as well as provide guidelines for extending the model to new deprivation conditions.

Response: We thank the reviewer for their positive feedback on the code repository and the web server implementation. The current version of the server is designed to be read-only, and does not support modeling with user-uploaded Ribo-seq data. Although the Python PIP package that we have provided for Riboclette can be readily installed to predict ribosome profiles and to interpret attributions for custom datasets. As suggested, the requirements to locally deploy the server have now been mentioned in the GitHub repository. The proposed version of the Riboclette model cannot be extended to new deprivation conditions out of the box. The Riboclette model would have to be re-trained to account for any new deprivation conditions. For future versions of Riboclette, we plan on modifying it to make it easier to extend to new deprivation conditions.

Reviewer #3 (Remarks to the Author):

This paper introduces Riboclette, a conditional deep learning model designed to predict ribosome footprint profiles (RFPs) under amino acid deprivation conditions. The model features a dual-output architecture that predicts both control RFPs and the difference between deprived and control conditions, enhancing interpretability. Using transformer-based and BiLSTM backbones, Riboclette outperforms existing methods in predicting translation elongation dynamics. Integrated Gradients and in silico perturbation analyses reveal codon-specific stalling determinants, including poly-basic regions and deprived amino acids.

Response: We thank the reviewer for their detailed analysis of the manuscript and for their positive feedback on the performance of the proposed Riboclette model.

Although the experiments exhibit encouraging results, I have some concerns as below:

1.The model architecture is not clearly presented in the paper. Figure 1b only illustrates the overall framework, but the dual-output heads mentioned in the text are not depicted. This aspect requires further elaboration.

Response: We thank the reviewer for their suggestion to outline clearly the double head architecture of the Riboclette model. We have now edited **Fig. 1B.** with clear labels for the dual-output heads (“Control Head” and “Depr. Diff. Head”), and we outline these in red to increase figure clarity.

2.In my opinion, the term "conditional" in this study refers merely to incorporating an amino acid deprivation as an additional input, rather than the model itself formulating a conditional structure. Therefore, the term "conditional deep learning model" might be misleading.

Response: We understand the reviewer’s concern regarding the usage of the term “conditional” to describe the nature of the Riboclette framework. Our aim was only to convey

that the predicted RFP was conditioned on the token that describes the amino acid deprivation. We have now edited the text (**Section 2.1**) to explain this more clearly.

3. Another significant shortcoming of this work is the lack of comprehensive performance validation. The authors only consider Riboformer with RiboMIMO as baseline, neglecting recent works such as those presented in DOI: 10.1126/sciadv.ado0738 and DOI: 10.1038/s42256-024-00915-6.

Response: We thank the reviewer for their comment to compare the proposed Riboclette framework with other frameworks such as Riboformer [DOI: <https://doi.org/10.1038/s41467-024-46241-8>], EIF [DOI: 10.1126/sciadv.ado0738], and Translatomer [DOI: <https://doi.org/10.1038/s42256-024-00915-6>]. We haven't done so as we believe that these methods are quite different in scope as compared to Riboclette. The primary aim of our manuscript is to study the RFPs in different amino acid deprivation conditions using only the full mRNA sequence. Below, we summarize the key differences in the scope and architecture between Riboclette and the suggested methods:

- Riboformer applies an mRNA window-based approach and also requires both the mRNA sequence and starting RFP counts to predict the target RFP counts. Although they discuss a sequence-only variant that could be used for comparison [DOI: <https://doi.org/10.1038/s41467-024-46241-8>: **Supp Table 4**], the code for this is not publicly available. As we discuss extensively in the answer to the first remark by R2, there are also significant differences between their motif extraction pipeline and ours.
- The EIF model is an anomaly detection framework that can only identify if a certain codon position is an RFP peak or not. It cannot be employed to predict full RFPs for given genes; hence, it cannot be used to conduct a comparison.
- Translatomer is capable of predicting the RFPs for a given gene, but it uses RNA-Seq in addition to the mRNA sequence to make the prediction. Comparing these two models would not be fair, as Riboclette is not designed for such a multimodal scenario. We also remark that the two analyses have a different scope, as RNA-seq and Ribo-seq are highly correlated modalities, while we are trying to exploit and interpret the information encoded only in the sequence.

We add more information about our motivation behind choosing the baseline models to conduct the performance comparison in **Section 2.1**.

4. The study relies only on Pearson Correlation Coefficient (PCC) for performance evaluation, which may not be sufficiently persuasive. One metric often reflects only one aspect of a model's predictive capability, it is recommended to include additional metrics such as R^2 or RMSE to provide a more comprehensive assessment.

Response: We agree with the reviewer's concern about providing multiple metrics to conduct the performance evaluation. We have already done so by evaluating the model performances using the Mean Arctangent Absolute Percentage Error (MAAPE) (**Fig. A2**). We appreciate the suggestion to use either R^2 or RMSE, but we believe that the MAAPE metric is a better fit for our use case as it is more robust to inputs of lower magnitude.

5. The attribution analysis based on integrated gradients, along with the subsequent experiments on model interpretability, is interesting. Would it be possible to perform attribution analysis using self-attention weights within the model? This could further demonstrate the model's interpretability.

Response: We thank the reviewer for their suggestion to further demonstrate the model's interpretability by considering the self-attention weights of the transformer model. Although we find this approach to be quite interesting, we believe that the interpretations from the self-attention weights would be far noisier and less reliable compared to those from integrated gradients [DOI: [10.48550/arXiv.1902.10186](https://doi.org/10.48550/arXiv.1902.10186)]. Considering that the gradient-based approaches are believed to be more reliable, we have not conducted any attention-based interpretability analyses.

6. I wonder if the amino acid starvation-specific codons and codon motifs identified through attribution analysis are significantly different from those obtained via statistical correlation?

Response: We thank the reviewer for their suggestion to compare the outputs of the proposed motif identification pipeline with those obtained from statistical correlations. We conduct a Fisher test of statistical significance to measure the enrichment of the identified motifs in the neighborhood of RFP peaks (protocol described in **Section 4.9** and results displayed in **Fig. 3C**). Motifs of length 1 and 2 are more often found to be identified through the statistical test as well, but not the motifs of length 3 which could potentially represent novel findings from the beam search pipeline that were missed by simple statistical significance tests.

Reviewer #3 (Remarks on code availability):

The GitHub link provided by the authors is accessible, and I can see the code, but there are no readily available data files to reproduce the work immediately.

Response: All the files related to the processed dataset, model predictions, and model interpretability outputs have already been provided in the dataset link in the GitHub repository (<https://os.unil.cloud.switch.ch/swift/v1/lts2-riboclette/>). These can be readily downloaded to easily run the Riboclette pipelines.

We thank the reviewers for evaluating our revised manuscript and for providing additional feedback to further improve it. We appreciate the positive remarks regarding the additions made to the manuscript and the clarifications provided in response to their earlier comments. Below, we address each of the reviewers' comments individually.

Reviewers' comments:

Reviewer #2 (Remarks to the Author):

Reviewer #1 and I shared similar concerns regarding the model's generalizability and the lack of motif validation. The authors have provided computational validation for their motif analysis. As for the model's limited scope and generalizability, the authors offered a detailed explanation, attributing it to the scarcity of datasets under diverse amino acid deprivation conditions. While this issue has not been fully addressed, I think the authors' explanation is reasonable, and the model itself may offer significant value to the field.

The authors have also addressed my other concerns.

I recommend this manuscript for publication in Communications Biology.

Response: We thank the reviewer for their positive feedback and for their recommendation of this manuscript for publication.

Reviewer #3 (Remarks to the Author):

1. Although the author provided point-by-point responses to my questions, the revisions to the content are limited. In particular, the performance comparison with existing methods remains insufficient.

Response: We appreciate the reviewer's suggestion to compare our proposed method with other existing approaches. Upon consideration, we agree that incorporating these additional methods could strengthen the manuscript and provide more value to the readers. Hence, we compared Riboclette with two more existing methods, Riboformer and iXnos, in addition to the already mentioned RiboMIMO baseline. Both Riboformer and iXnos use codon-windows as the input and predict the ribosome profile counts for that specific A-site, whereas RiboMIMO and Riboclette use the full mRNA CDS as the input to predict the entire ribosome footprint profile. For Riboformer, we implement a sequence-only variant that does not depend on the starting condition ribosome profiling counts in order to conduct a fair comparison between the methods. Given that two of the baselines, Riboformer and iXnos, truncate the input CDS at the termini during data processing, we standardize this for every method by applying the more extensive Riboformer approach of truncating 20 codons at both ends of the sequence. As a result, we updated the prior results based on the new pre-processing. Considering that these baseline models do not account for the deprivation condition, different instances of them were trained for each of the six conditions.

Through this benchmarking study, we notice that Riboclette attains the highest performance in accurately predicting the ribosome footprint profiles across different amino acid

deprivation conditions. The updated results for the benchmarking study for the different metrics can be found in Figures 1 (e, f), A2, and A3. We update the methods section 4.5 (Baselines) to mention the two new baseline methods (line numbers 510-520), and results section 2.1 (Transformer-based Riboclette is the best performing model across conditions) to mention that Riboclette outperforms all three baseline models (line numbers 126-157).

2. For my concern on sufficient performance metric, the author supplement only one additional performance metric—Mean Arc-tangent Absolute Percentage Error (MAAPE), in Fig. A2. I still suggest to include other more evaluation metrics.

Response: We thank the reviewer for highlighting the need for more performance metrics. We now provide two more metrics, Mean Squared Error (MSE) and Spearman Correlation Coefficient (SCC), in addition to the previously mentioned Pearson Correlation Coefficient (PCC) and Mean Arc-tangent Absolute Percentage Error (MAAPE) metrics. The methods section 4.7 (Model Training and Evaluation) has been updated accordingly to reflect this (line numbers 536-539).

3. The revised content were not highlighted, making it difficult to identify which parts have been modified by the author.

Response: We believe that we have included a LaTeX difference file highlighting the changes made to the manuscript. However, we apologize for any miscommunication should this file not have been received. For this submission, we do the same and attach the latest version of the LaTeX difference file, highlighting the differences and edits made to the manuscript. The list of documents included in this submission is:

- Edited manuscript file
- LaTeX difference between the original manuscript and the resubmitted manuscript
- LaTeX difference between the resubmitted manuscript and the last version
- Appendix Table A3 with the motif statistics.